# Biochemical characterization and inhibition of the alternative oxidase enzyme from the fungal phytopathogen *Moniliophthora perniciosa*

Mario R. O. Barsottini [1,2], Alice Copsey[2], Luke Young [2], Renata M. Baroni [1], Artur T. Cordeiro[3], Gonçalo A. G. Pereira [1]✉ & Anthony L. Moore[2]✉

*Moniliophthora perniciosa* is a fungal pathogen and causal agent of the witches' broom disease of cocoa, a threat to the chocolate industry and to the economic and social security in cocoa-planting countries. The membrane-bound enzyme alternative oxidase (MpAOX) is crucial for pathogen survival; however a lack of information on the biochemical properties of MpAOX hinders the development of novel fungicides. In this study, we purified and characterised recombinant MpAOX in dose-response assays with activators and inhibitors, followed by a kinetic characterization both in an aqueous environment and in physiologically-relevant proteoliposomes. We present structure-activity relationships of AOX inhibitors such as colletochlorin B and analogues which, aided by an MpAOX structural model, indicates key residues for protein-inhibitor interaction. We also discuss the importance of the correct hydrophobic environment for MpAOX enzymatic activity. We envisage that such results will guide the future development of AOX-targeting antifungal agents against *M. perniciosa*, an important outcome for the chocolate industry.

[1] Genomics and bioEnergy Laboratory, Institute of Biology, University of Campinas, Campinas, Brazil. [2] Biochemistry & Biomedicine, School of Life Sciences, University of Sussex, Brighton BN1 9QG, UK. [3] Brazilian Biosciences National Laboratory, Brazilian Center for Research in Energy and Materials, Campinas, Brazil. ✉email: goncalo@unicamp.br; a.l.moore@sussex.ac.uk

The cocoa tree (*Theobroma cacao*) is a perennial plant originated from the Amazon River basin in Central America. Cocoa beans are the main raw material used in the multibillion-dollar chocolate industry and the cocoa pod is used in cosmetics, food thickeners, beverages, and others. On the other hand, fungal pathogens are one of the main causes of production losses in cocoa plantations, estimated at 30% worldwide on average, but affected farms can be completely ruined if left untreated. The basidiomycete fungus *Moniliophthora perniciosa* is the causal agent of the witches' broom disease of cocoa (WBD) in Latin America, and a WBD outbreak in Brazil's main cocoa-producing region—the state of Bahia—in 1989 led to a substantial loss in Brazilian competitiveness. Brazil's cocoa production fell by 75%, and Brazil has consequently shifted from a net exporter to a net importer of cocoa beans, resulting in an economic, social and environmental crisis[1–3].

*M. perniciosa* is a hemibiothtophic pathogen with a complex lifestyle and two physiologically distinct infective stages. The infection begins when *M. perniciosa* basidiospores reach meristematic cocoa tissues, whereupon infective hyphae penetrate through stomata and wounds. Initially, *M. perniciosa* develops without killing its host, which is known as the biothrophic stage of WBD[2]. It is evident that *M. perniciosa* employs effective molecular tools to suppress or circumvent the plant defence responses and of particular interest is the identification of an alternative oxidase (AOX)-coding gene. We have shown that AOX is highly expressed during the *M. perniciosa*–cocoa interaction and is critical for disease progression[4,5].

There is substantial evidence that the AOX, a stress-induced respiratory chain protein, ensures a degree of metabolic plasticity to the cell, both during regular metabolism and stressful conditions, such as the inhibition of complexes III or IV, which halts cellular respiration and ATP production. Such oxidative imbalance also leads to the generation of harmful reactive oxygen species. AOX, however, acts as a bypass to complexes III and IV, thereby partially maintaining ATP synthesis through enhanced complex I activity[6].

Infected cocoa plants produce nitric oxide, a potent complex IV inhibitor, and we have demonstrated that chemical inhibition of AOX prevents WBD progression[5,7]. Furthermore, *M. perniciosa* AOX (MpAOX) is a key resistance factor to commercial fungicides, such as strobilurins, complex III inhibitors[5], and the same has been observed for other fungal plant pathogens which threaten global food security, such as *Botrytis cinerea*, *Magnaporthe grisea*, *Mycosphaerella graminicola*, *Venturia inaequalis* and others[8,9]. This clearly makes AOX a desirable target for the development of novel crop protection agents, however currently available AOX inhibitors are not suitable for commercial use. For instance, salicylhydroxamic acid (SHAM) and *n*-propyl gallate inhibit *M. perniciosa* growth only at millimolar concentrations[5,7]. Colletochlorin B and structurally related compounds, such as ascofuranone and ascochlorin comprise a novel class of AOX inhibitors with potential use as antimicrobial agents[10,11]. Nonetheless, AOXs from different organisms display varying degrees of sensitivity to those inhibitors owing to differences in the ligand-binding site[12], which can severely affect the outcome of a treatment against WBD.

The study of fungal AOX has focused primarily on the interplay between the main and alternative respiration pathways throughout their life cycle, during pathogenesis or in response to stress[13]. In general, AOX is induced after treatment with main respiration inhibitors or reactive oxygen species (ROS)-generating compounds, as seen in *Ustilago maydis*[14], *Sclerotinia sclerotiorum*[15] and *Aspergillus fumigatus*[16]. AOX expression may increase during cell morphology changes, such as the development of biotrophic hyphae in *M. perniciosa*[5], microsclerotia

formation in *Nomuraea rileyi*[17] and the transition from yeast-like to filamentous (infective) forms in the opportunistic human pathogens *Candida albicans* and *Paracoccidioides brasiliensis*[18,19]. Interestingly, genetic ablation of AOX activity has led to reduced virulence or survival rates in *Cryptococcus neoformans* and *A. fumigatus*[20,21]. Finally, *Aspergillus nidulans* and *A. fumigatus* exhibit a correlation between higher AOX activity and higher production of secondary metabolites, such as sterigmatocystin and cephalosporin C[22,23].

The heterologous expression of AOX in eukaryotic or prokaryotic cells has been useful to investigate biochemical features of the enzyme, such as resistance to traditional main respiratory inhibitors, ROS-scavenging activity, determination of the substrate-binding site and post-translational regulation of AOX[24–32]. To our knowledge, works pertaining to the development of novel AOX inhibitors are scarce with the exception of the human parasite *Trypanosoma brucei*—the best-studied AOX in that regard and the only one for which a crystal structure is available—and a few others, such as *Trypanosoma vivax*, *Antonospora locustae* and *Blastocystis hominis*[10,33–38]. Moreover, because AOX is a membrane-bound protein, it is usually purified and biochemically characterised in the presence of detergents which may not accurately reflect the phospholipid bilayer biophysical properties and are not compatible with the natural AOX substrate—ubiquinol-10 ($Q_{10}H_2$)[29,33,34].

The goal of this study was to kinetically characterise MpAOX and determine its sensitivity to CB and analogues in order to develop specific novel antifungals targeted at *M. perniciosa*. To that end, we have heterologously expressed and purified the recombinant MpAOX (rMpAOX) and demonstrate that it is active in *Escherichia coli* membranes. This system was used to evaluate putative AOX activators and putative AOX inhibitors in dose–response assays. The analysis of an MpAOX structural model and comparison with the known structure of *T. brucei* AOX (TAO) allowed the identification of key amino acid residues that influence protein-ligand affinity. Finally, we present and discuss the kinetic characterisation of the purified rMpAOX before and after incorporation into a self-assembled proteoliposome system that better reflects the natural hydrophobic phospholipid membrane environment.

## Results

**Membrane-bound rMpAOX**. The rMpAOX construct used in this work corresponds to the mature MpAOX enzyme without the predicted mitochondrial leading peptide (first 40 N-terminal amino acid residues), which is cleaved following import into the mitochondrial matrix[39]. The removal of the mitochondrial leading sequence increases recombinant TAO stability[35].

Initially, membrane-bound rMpAOX activity was measured using a high-resolution oxygraph. Fungal AOXs are activated by mono- and diphosphate nucleosides, such as AMP, GMP and ADP, whereas plant AOXs are activated by organic acids, such as pyruvate, succinate and others[40,41]. As can be seen from Fig. 1a, 1 mM GMP increased membrane-bound rMpAOX oxygen uptake 9.8 times, to a maximum of 0.67 μmol $O_2$ min$^{-1}$ mg total protein$^{-1}$ and 5 μmol $O_2$ min$^{-1}$ mg AOX$^{-1}$ (Supplementary Fig. 1). AMP and ADP stimulated rMpAOX by 8.1 times and 4.5 times, respectively. A dose–response assay with AMP, ADP and GMP reveals that AMP activates rMpAOX as much as GMP at higher concentrations, whereas 6.5 mM ADP was not enough to sustain maximum rMpAOX rates. EC$_{50}$ values for GMP, AMP and ADP are 166 μM, 918 μM and ca. 6.5 mM (Fig. 1b). Interestingly, GDP did not have any noticeable effect on rMpAOX at 1 mM (Fig. 1a) and at 6.5 mM it increased rMpAOX basal rate by only 1.5 times (Supplementary Fig. 2).

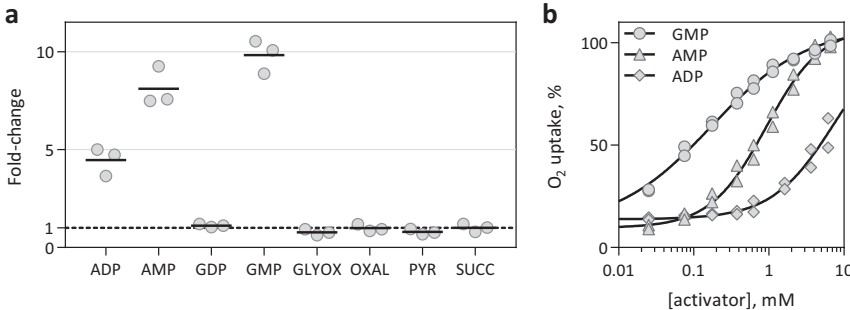

**Fig. 1 Testing of rMpAOX activators. a** Fold-change in membrane-bound rMpAOX $O_2$ uptake after addition of putative activators (1 mM each): ADP, AMP, glyoxylate, GMP, oxaloacetate, pyruvate and succinate. The dashed line indicates a fold-change of 1 (e.g., no change after addition). GMP caused the greatest increase in enzymatic activity. Horizontal bars indicate the mean of three independent measurements (circles). **b** Dose–response assay with GMP. Rates were normalised with respect to the maximal rate achieved. Points represent two independent measurements and the continuous lines are the best fit for a 4-parameter sigmoid function. $EC_{50} = 166\,\mu M$ (GMP), 918 μM (AMP) and ca. 6.5 mM (ADP).

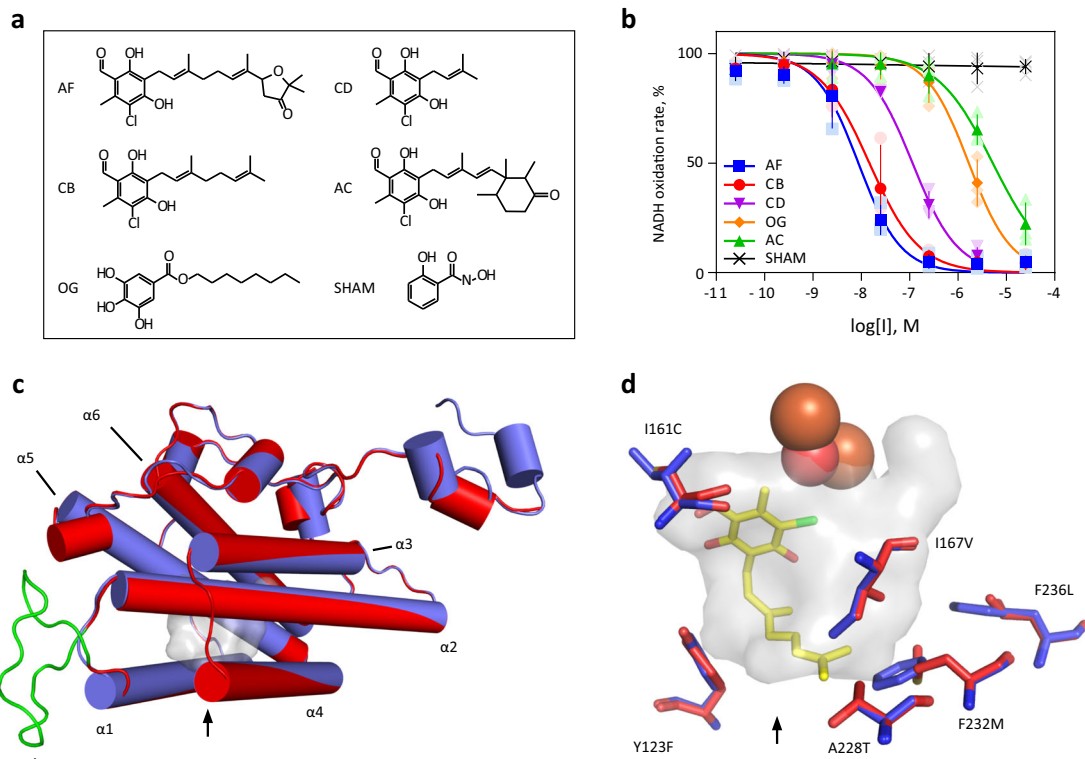

**Fig. 2 Structure–activity relationship of MpAOX inhibitors. a** Representation of ascofuranone (AF), colletochorin B (CB), colletochlorin D (CD), ascochlorin (AC), octyl gallate (OG) and salicylhydroxamic acid (SHAM). **b** Dose–response assay with membrane-bound rMpAOX. Symbols represent the mean and standard deviation of three independent measurements. The continuous lines are the best fit of a 4-parameter sigmoid function used to determine $IC_{50}$ values which are follows. AF: 8.8 nM, CB: 14.8 nM, CD: 115 nM, OG: 1,673 nM and AC: 5,246 nM. **c** Overview of MpAOX (red; computer-generated) and TAO (blue; crystal structure PBB ID 3w54) structures. The asterisk marks an insertion of 18 residues in MpAOX (green loop) found in fungal AOXs and related to GMP activation. The arrow indicates the entrance to the substrate-binding pocket (grey shading). **d** Comparison of MpAOX and TAO-binding pockets. Amino acid residues that differ between those enzymes are shown as red (MpAOX) and blue (TAO) sticks and listed in Table 1. MpAOX residue numbering is used as reference. Yellow sticks depict CB present in the TAO crystal structure, and orange and red spheres represent two Fe ions and one $OH^-$ ion in TAO.

Next, we evaluated four AOX inhibitors on membrane-bound, GMP-activated rMpAOX and compared the results to conventional AOX inhibitors such as salicylhydroxamic acid (SHAM) and octyl gallate (OG). As shown in Fig. 2b, SHAM did not exert any effect up to the maximal tested concentration of 25 μM, which is in accordance with previous results obtained with intact *E. coli* cells[7]. On the other hand, $IC_{50}$ values were successfully determined for ascochlorin (AC), ascofuranone (AF), colletochlorin B (CB), colletochlorin D (CD) and OG. Interestingly, AC,

AF, CB and CD possess identical head groups, allowing the effect of the distinct carbonic tail on inhibitor potency to be tested. Amongst those four analogues, AF and CB exhibited the lowest $IC_{50}$ values (8.8 and 14.8 nM, respectively), followed by CD (115 nM) and AC (5246 nM). OG $IC_{50}$ is 1673 nM.

Overall, the $IC_{50}$ values obtained with rMpAOX are higher than those reported for the membrane-bound TAO[10], which can be explained by differences in total AOX content. Most striking, however, was the broader range of inhibitor potency observed

with rMpAOX. For TAO, $IC_{50}$ values ranged from 0.2 nM (CB) to 1.5 nM (AC) with a ratio of 7.5; but this ratio is much greater for MpAOX and the AC $IC_{50}$ is 354 times larger than that of CB. This indicates that AC is a much weaker ligand for MpAOX—a valuable piece of information for the design of compounds targeted at *M. perniciosa*.

**MpAOX structural model**. In an attempt to gain further insights regarding the structure–activity relationships of MpAOX inhibitors, we compared MpAOX to TAO. Sequence alignment between both proteins revealed a complete conservation of amino acid residues involved in ligand binding[42] (Supplementary Fig. 3), which prompted us to create an MpAOX three-dimensional structural model (Fig. 2C) based on TAO. The MpAOX model is in good agreement with TAO crystal structure (PDB ID 3w54) with a Cα root mean square deviation of 1.6 Å. Importantly, the enzymatic core, which comprises six α-helices, which accommodate the hydrophobic substrate-binding site and the catalytic residues, is highly conserved. The main differences between the MpAOX model and TAO are seen in the flexible N-terminal region and in the extra loop in MpAOX (between α-helices α1 and α2), which corresponds to the insertion common to all fungal AOXs.

A comparison of amino acid residue composition, which make up the AOX ligand-binding cavity reveals six differences between MpAOX and TAO (Fig. 2D, Table 1). In general, the MpAOX residues are bulkier than the TAO counterparts rendering the MpAOX cavity smaller (165.89 and 225.72 Å³, respectively). A visual inspection of CB bound to TAO in the crystal structure shows that the end of the geranyl tail is close to residues V125 and M190, which in MpAOX correspond to I167 and F232, respectively. It is plausible that the bulky trimethylated cyclohexanone ring in AC clashes with those MpAOX residues and hence explains the large drop in potency observed with this inhibitor. Interestingly, the furanone ring in AF is not detrimental for MpAOX-ligand interaction, suggesting that the longer geranyl tail of AF (8C atoms in comparison to 6C in AC) is sufficient to prevent a clash with I167 and F232.

The activation of fungal AOXs by mono- and dinucleotides seems to be a direct result of an insertion of ~20 amino acid residues in those proteins[40]. In our structural model, this insertion is found between α-helices α1 and α2 (Fig. 2c) and is depicted as a disordered loop possibly due to a lack of homology with the template protein (TAO). However, several secondary structure prediction algorithms indicate that the fungal insertion would be exposed to the solvent and adopt a helical conformation (Supplementary Fig. 4). Such ordered structure could form a binding site for allosteric regulation of fungal AOXs.

**rMpAOX purification and reincorporation into proteoliposomes**. AOX is a protein that naturally attaches to hydrophobic phospholipid membranes, hence the purification in aqueous medium is carried out with surfactants (detergents). An initial

**Table 1 Differences in amino acid composition in the substrate-binding pocket between MpAOX and TAO.**

| MpAOX | TAO |
|---|---|
| Y123 | F99 |
| I161 | C119 |
| I167 | V125 |
| A228 | T186 |
| F232 | M190 |
| F236 | L194 |

screening with various detergents indicated that DDM and FC-12 were the most efficient for membrane removal whilst maintaining rMpAOX activity and those detergents were selected for further evaluation. C12E8, which improved TAO activity when included in the enzyme reaction buffer[33], was also tested. As seen in Fig. 3a, the use of 1% FC-12 during protein solubilisation yielded the highest rMpAOX enzymatic activity regardless of the detergent subsequently used for purification. Based on these results, we selected 1% FC-12 for membrane protein solubilisation and 0.05% DDM (~3× the critical micellar concentration) for downstream steps. Therefore, a highly pure protein of the expected size for rMpAOX was obtained (Fig. 3b).

Purified rMpAOX was characterised with respect to the hydrophilic substrate ubiquinol-1 ($Q_1H_2$). A titration of C12E8 in the reaction buffer showed that 0.0125% resulted in maximal enzymatic activity with an increase of 3.5 times the activity when compared to the non-detergent condition and was used in all subsequent $Q_1H_2$ assays (Supplementary Fig. 5). The $K_M$ and $V_{Max}$ for rMpAOX were therefore determined as $68.5 \pm 8.8\ \mu M$ and $504.1 \pm 28.7$ nmol $Q_1$ min$^{-1}$ mg$^{-1}$, respectively (Fig. 3c, Supplementary Fig. 6A). Further addition of GMP up to 0.5 mM did not alter the rMpAOX kinetic parameters such as $K_M$ and $V_{Max}$. Moreover, the CB $IC_{50}$ on the purified rMpAOX was determined as 59.3 nM (Fig. 3d). As the $IC_{50}$ value is within a 10-fold difference with respect to the enzyme concentration in the assay (49.1 nM, yielding a CB-to-rMpAOX ratio of 1.2), we used the Morrison equation for calculating the $K_{Iapp}$ of tight-binders, which gave an $K_{Iapp}$ for CB of 13.9 nM[43]. Note that CB is a mixed-type AOX inhibitor and therefore $K_{Iapp}$ is a function of $K_I$ and $K_I'$ parameters[33,36].

In addition to the detergent-solubilised rMpAOX, we employed a recently developed proteoliposome system to characterise rMpAOX that better reflects the hydrophobic environment of the phospholipid membrane[44]. This also allowed us to test the natural quinol analogue found in mitochondria, namely ubiquinol-10 ($Q_{10}H_2$), which is too hydrophobic to be used in aqueous solution. Therefore, we prepared self-assembled proteoliposomes (PLs) containing rMpAOX, $Q_{10}$ and a bacterial NADH:ubiquinone oxidoreductase (NDH-2) as an electron donor to $Q_{10}$. A range of $Q_{10}$ concentrations up to 17 mM was tested, which allowed the determination of kinetic parameters of membrane-bound rMpAOX, resulting in $K_M = 0.96 \pm 0.76$ mM and $V_{Max} = 59.4 \pm 6.6\ \mu mol\ Q_{10}$ min$^{-1}$ mg$^{-1}$ (Fig. 4, Supplementary Fig. 6B). The CB $IC_{50}$ in our PLM preparation was 19.5 nM, which is similar to values reported for TAO in a similar defined PL system using complex I as the electron donor[45].

## Discussion

Chocolate is appreciated worldwide and its industry grows by the year with the top 10 chocolate manufacturers reaching net sales of $78 bn in 2018. Cocoa beans are essential for chocolate production but cocoa farms in Latin America are threatened by *M. perniciosa*. *M. perniciosa*, in addition to other pests and diseases, can decimate farms if left untreated, and there is a substantial risk of this pathogen reaching other cocoa-producing countries in Africa and Asia with disastrous consequences[46]. We have demonstrated that a class of compounds initially identified as TAO inhibitors are also effective against MpAOX. Such information is valuable for the future design of antifungal agents specifically targeted at *M. perniciosa*.

Of particular importance was the observation that AC was less potent against MpAOX than observed for TAO, which led to insights regarding structure–activity relationships of CB analogues on MpAOX. The identification of features that confer inhibitor specificity between different AOXs is especially

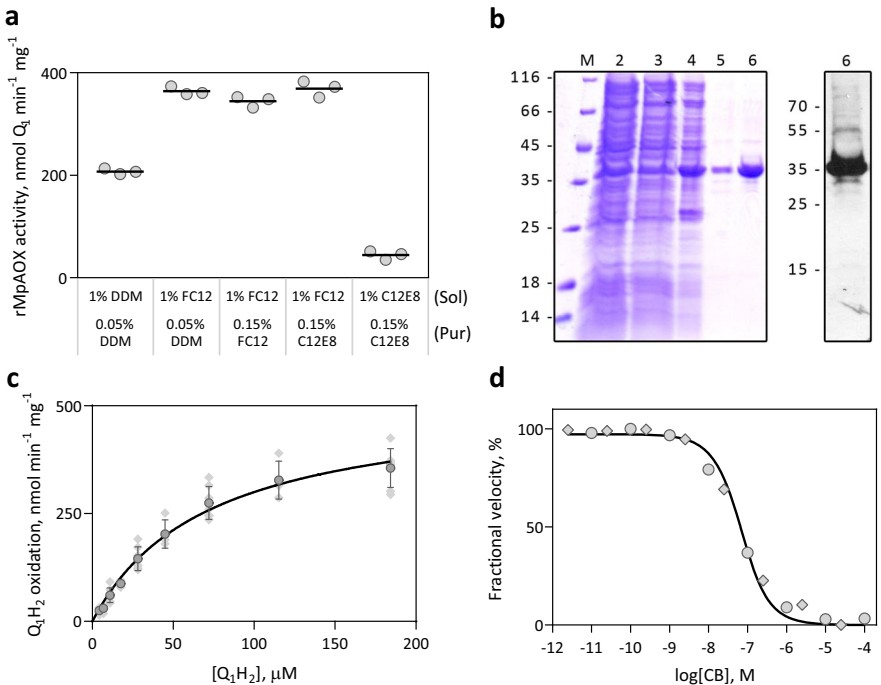

**Fig. 3 Purification and kinetic characterisation of solubilised rMpAOX. a** Effect of detergents DDM, FC-12 and C12E8 on pure rMpAOX activity. Horizontal bars depict the $Q_1H_2$ oxidation rate of rMpAOX purified with the indicated detergents during membrane protein solubilisation (top label) and affinity chromatography (bottom label). Measurement of three technical replicates. Membrane protein solubilisation with FC-12 allowed for the highest rMpAOX activity regardless of the detergent used downstream. **b** Coomassie brilliant-blue-stained SDS-PAGE (left) and western blot with anti-AOX antibody (right) of protein samples collected during rMpAOX purification. M: molecular weight marker; 2: total *E. coli* protein extract; 3: clarified cell lysate; 4: *E. coli* membrane faction; 5: affinity chromatography flow-through; 6: affinity chromatography eluate. **c** Kinetic characterisation of rMpAOX with respect to ubiquinol-1 ($Q_1H_2$). The assay was performed in 50 μL 20 mM Tris-HCl buffer with 0.0125% C12E8 and 100 ng rMpAOX. Circles and error bars depict the mean and SD of seven independent measurements (diamonds). The continuous line is the best fit for the Michaelis–Menten kinetic model with $K_M =$ 68.5 μM and $V_{Max} = 504.1$ nmol $Q_1$ min$^{-1}$ mg$^{-1}$. **d** Dose–response assay with CB. Points represent two independent assays (circles or diamonds) and the continuous line is the best fit for the Morrison equation[43]. CB $IC_{50} = 59.3$ nM and $K_{Iapp} = 13.9$ nM.

important in agricultural settings where both plant and pathogen express AOX because—ideally, only the pathogen's enzyme should be inhibited to reduce the chances of side-effects on the plant. It has been previously demonstrated that CB analogues may show distinct inhibitory profiles between different AOXs, and now we present a mechanistic explanation for such an observation[12]. Specifically, differences in the entrance of the hydrophobic channel, which leads to the catalytic core may alter inhibitor affinity. MpAOX amino acid residues I167 and F232 clash with the AC trisubstituted hexanone ring and prevent favourable protein–ligand interactions. This in turn can be avoided by increasing the linker length, as seen with AF or avoiding bulky substituents such as in CB. In other words, the length of the aliphatic tail may not only directly modulate the AOX–ligand interaction through the number of contact points made in the hydrophobic tunnel—as shown for TAO[10,35,47]—but also indirectly through the positioning of large substituents at the tunnel entrance.

Inhibitor potency is an important parameter for drug development projects. Derivatives of salicylic acid and of gallic acid are well-known AOX inhibitors[48–50], but are only effective against *M. perniciosa* at millimolar concentrations[5,7]. We have also recently shown that another AOX inhibitor, *N*-(3-bromophenyl)-3-fluor-obenzamide, has a protective effect in plants against WBD at 200 μM even in the absence of a main respiration inhibitor[7]. The high concentrations of molecules necessary to inhibit fungal growth are not ideal from an agricultural and commercial perspective. Therefore, we anticipate that AF and CB analogues, which are tight-binding inhibitors, will lead to more effective antifungal agents against *M. perniciosa* and other fungal pathogens.

We observed a drop in rMpAOX enzymatic activity after removal of the bacterial membrane from 5 μmol $O_2$ min$^{-1}$ mg AOX$^{-1}$ (equivalent to 10 μmol Q min$^{-1}$ mg AOX$^{-1}$) to a $V_{Max}$ of 504.1 nmol $Q_1$ min$^{-1}$ mg$^{-1}$ with the solubilised rMpAOX. This is opposed to results from TAO and SgAOX for which the specific activity increases after purification, which prompted us to investigate the dependence of rMpAOX activity on the phospholipid membrane[29,33]. Therefore, rMpAOX was incorporated into defined PLs with $Q_{10}$, the natural substrate present in mitochondria and allowed rMpAOX to reach a considerably higher rate $V_{Max}$ of 59.4 μmol $Q_{10}$ min$^{-1}$ mg$^{-1}$. In fact, the rMpAOX catalytic rate with 180 μM $Q_{10}H_2$ in the phospholipid membrane is calculated to be 9.3 μmol min$^{-1}$ mg$^{-1}$ based on $K_M$ and $V_{Max}$ values, which is 18.4 times higher than observed for $Q_1H_2$.

The difference between rMpAOX catalytic rate before and after PL assembly might have been explained by substrate availability. The low solubility of $Q_1H_2$ in aqueous medium may hinder AOX kinetic characterisation, which can be overcome with the use of detergents to prevent $Q_1H_2$ micelle formation at concentrations up to 600 μM[33,51]. Although we have included C12E8 in the enzymatic reaction medium for the pure rMpOAX, we did not immediately discard the possibility of $Q_1H_2$ aggregation, which may explain the lower rMpAOX rates after solubilisation due to differences in our experimental setup compared with previous experiments. For instance, Hoefnagel et al.[51] describe that micelle formation is time-dependent and our reaction in microtiter plates might take longer to prepare and read than assays previously performed in quartz cuvettes. In order to investigate this

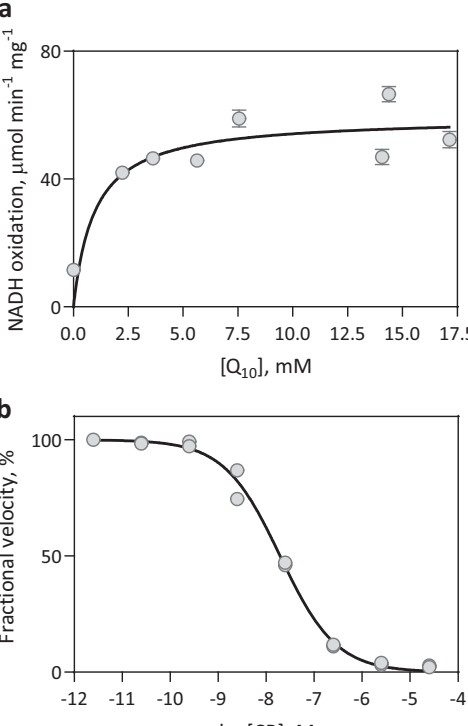

**Fig. 4 Characterisation of rMpAOX incorporated into proteoliposomes. a** Kinetic characterisation of rMpAOX with respect to ubiquinol-10 ($Q_{10}H_2$). The assay was performed in 200 μL 65 mM MOPS pH 7.5 with a starting concentration of 300 μM NADH and ~50 ng rMpAOX. Each point represents one PL preparation for which a different amount of $Q_{10}$ was added with the SD of technical triplicates. Protein and $Q_{10}$ quantifications were performed as described in ref. [44]. The continuous line is the best fit for the Michaelis–Menten kinetic model. $K_M = 0.96 \pm 0.76$ mM and $V_{Max} = 59.4 \pm 6.6$ μmol $Q_{10}$ min$^{-1}$ mg$^{-1}$. **b** Dose–response assay with CB. Points represent two independent measurements and the continuous line is the best fit for a four-parameter sigmoid function. CB $IC_{50} = 19.5$ nM.

hypothesis, we used $Q_1H_2$ as substrate in PL assays, which yielded essentially the same result as the detergent-solubilised rMpAOX (Supplementary Fig. 7) demonstrating that the phospholipid membrane is not sufficient to sustain maximal rMpAOX activity with $Q_1H_2$. This is in agreement with results obtained with complex I showing that, even in the PL system, the ubiquinone isoprenoid tail length influences the enzyme turn-over rate with $Q_1$ and $Q_2$ yielding lower rates than $Q_4$, $Q_6$, $Q_8$ and $Q_{10}$[45]. In fact, structural features of the AOX quinol-binding site and biochemical data obtained with inhibitors (e.g., compare CB and CD) have shown that molecules with short lipophilic tails interact less effectively with AOX[6,33,42]. Furthermore, CB dose–response assays indicate that the rMpAOX–ligand interaction (and therefore the rMpAOX ligand-binding site) was not disturbed after solubilisation. The CB-to-rMpAOX ratio at the $IC_{50}$ concentration is 1.2 and similar values have been reported for solubilised TAO (1.3) and for TAO incorporated into PLs (1.4)[34,45]. Overall, those results show that the lower activity of the solubilised rMpAOX is not due to $Q_1H_2$ solubility issues and is instead caused by a weaker protein-substrate interaction. This highlights one advantage of using the PL system for AOX characterisation— namely the ability to use the physiological substrate at controlled concentrations up to saturation where it would otherwise prove problematic in aqueous solution. Another advantage of the PL system is the fact that AOX becomes the limiting step for electron flux thus reaching maximal catalytic rate, as evidenced by the fact

that in *E. coli* membranes rMpAOX is only ~16% as fast as in PLs (Fig. 4, Supplementary Fig. 1C). Most likely, the bacterial NADH dehydrogenase cannot provide sufficient amounts of reduced Q to sustain maximal activity of the overexpressed rMpAOX.

The effect of GMP on rMpAOX was also altered after removal of the bacterial membrane. The ectopically expressed enzyme retained the ability to be stimulated by GMP whilst attached to *E. coli* membranes, much like what is observed with other fungal AOXs in situ. For instance, the GMP and AMP $EC_{50}$ values for rMpAOX (166 and 918 μM) are in the same range of concentration necessary for half-maximal activation of *Hansenula anomala* mitochondria[52–55]. However, it is still unclear how variations of those nucleotides might regulate AOX activity in vivo under normal development and during stress response. The intracellular concentration of GMP, AMP and ADP are in the micromolar range in several cell types (bacteria, yeast and mammalian cells) suggesting that GMP has a greater role in AOX activation[56–58]. However, there might be spatial or temporal variations in the concentration of those nucleotides in mitochondria such as an increase in ADP levels caused by the impairment of the main respiratory chain. To our knowledge, there is no information with respect to variations in intramitochondrial GMP and AMP concentration.

No rMpAOX stimulation was observed with the purified protein even in the PL system and this prevented the determination of dissociation constants, $k_a$. The 18-amino acid fungal AOX insertion is close to α-helix 1 which forms—together with α4—the hydrophobic surface responsible for attachment to the phospholipid membrane (Fig. 2c). Hence, it is possible that the lipid environment is necessary for the correct conformation of the fungal loop or that an as-of-yet unknown molecular intermediate or accessory in the cell membrane is required for AOX stimulation. The sensitivity of rMpAOX to GMP was not regained in the PL system, which might indicate that either rMpAOX had already reached maximum catalytic rates or that the molecular intermediate found in mitochondria and in *E. coli* membranes is absent in our defined PLs.

## Methods

**Chemicals and reagents**. Ubiquinonol-1 ($Q_1H_2$) was obtained after the chemical reduction of ubiquinone-1 ($Q_1$; Sigma), as described elsewhere[59]. In summary, 10 mg $Q_1$ were dissolved in 5 mL ethanol, to which 4 g $Na_2S_2O_4$ in 30 mL of water were added. The reaction was left for 1 h in the dark and $Q_1H_2$ was extracted with *n*-hexane (3 × 20 mL). The pooled *n*-hexane fractions were washed with 20 mL water, dried with solid $MgSO_4$ and filtered. The *n*-hexane was removed by evaporation and the solid residue was resuspended with $CDCl_3$ for nuclear magnetic resonance spectroscopy to confirm the identity and purity of $Q_1H_2$. The chloroform was removed by evaporation and $Q_1H_2$ was dissolved with acidified DMSO and stored at −80 °C until use. To determine the $Q_1H_2$ concentration, the stock solution was diluted 100-fold in water and left to spontaneously oxidise for 24 h, after which the absorbance at 278 nm was determined ($\varepsilon_{278} = 15$ mM$^{-1}$ cm$^{-1}$)[33]. Detergents were obtained from Anatrace and putative AOX activators from Sigma.

**rMpAOX cloning and expression**. The MpAOX sequence used in this work corresponds to UniProt ID A8QJP8[5], which was cloned without the predicted mitochondrial leading sequence, as determined with the Signal P server[60]. Strep-tagged rMpAOX was synthesised and cloned into the pET15a (Genscript). rMpAOX expression, *E. coli* membrane fractionation and protein purification were performed essentially as described for *Sauromatum guttatum* AOX[29,33]. FN102 *E. coli* cells transformed with pET15b:rMpAOX plasmid were inoculated into 10 mL of L-broth supplemented with 100 μg mL$^{-1}$ ampicillin, 100 μg mL$^{-1}$ kanamycin and 50 μg mL$^{-1}$ 5-aminolevulinic acid (ALA). This was left to grow overnight at 37 °C. On the following day, 1 mL of overnight culture was transferred to 50 mL of L-broth supplemented with 100 μg mL$^{-1}$ ampicillin, 100 μg mL$^{-1}$ kanamycin, 50 μg mL$^{-1}$ ALA, 0.2 % (w/v) glucose, 50 μg mL$^{-1}$ $MgSO_4$, 25 μg mL$^{-1}$ $FeSO_4$, and 25 μg mL$^{-1}$ $FeCl_3$. The starter culture was left at 37 °C in a shaking incubator (180 rpm) until the $OD_{600}$ reached 0.6. The cells were harvested by centrifugation, resuspended with ~5 mL of fresh K-broth and inoculated in 4 × 1 L of K-broth supplemented with 100 μg mL$^{-1}$ carbenicilin, 100 μg mL$^{-1}$ kanamycin, 50 μg mL$^{-1}$ $MgSO_4$, 25 μg mL$^{-1}$ $FeSO_4$, 25 μg mL$^{-1}$ $FeCl_3$ and 0.2 % (w/v) glucose, with an starting $OD_{600}$ of 0.01. The culture was left at 30 °C with 180 rpm shaking until the OD reached ~0.6, then

50 μM isopropyl β-D-1-thiogalactopyranoside (IPTG) was added for induction of rMpAOX expression. Cultures were left at 30 °C for 14-16 hours before cell harvesting.

**E. coli membrane fraction preparation**. *E. coli* membrane preparation was performed as described with minor modifications[29]. The cell pellet was thoroughly resuspended in 10 mL of 65 mM MOPS buffer pH 7.5 per gram of cells supplemented with protease inhibitor tablets (Roche; 1 tablet per 50 mL), 2.5 U mL$^{-1}$ of benzonase (Sigma) and 100 mM MgSO$_4$. The cells were disrupted using two passes at 30 kPa through a pre-cooled Constant cell disruption system (Constant Systems Ltd). The lysate was centrifuged for 15 min at 16,000 rcf, and the pelleted unbroken cells and large cell debris were discarded. The supernatant was centrifuged for 90 min at 200,000 rcf to collect the fragmented membranes. After centrifugation the supernatant was discarded and the membrane pellets were thoroughly resuspended with 65 mM MOPS pH 7.5 with the aid of a Dounce homogenizer.

**rMpAOX purification**. The *E. coli* membrane fraction was resuspended with 50 mM Tris-HCl pH 7.5, 20% glycerol, 200 mM MgSO$_4$ and 1% Fos-Choline-12 (FC-12) instead of MOPS buffer. The sample was left under gentle rocking for 1 h at 4 °C, after which it was centrifuged at 200,000 rcf for 30 min. The supernatant was then subjected to affinity chromatography with a 5-mL StrepTrap HP column (GE Healthcare) in an Äkta FPLC (GE Healthcare) at 4 °C. The column was equilibrated with 65 mM MOPS pH 7.5 before passing the sample, after which the resin was washed with 10 volumes of buffer A (65 mM Tris-HCl pH 7.5, 50 mM MgSO$_4$, 60 mM NaCl, 20% glycerol and 0.05% DDM, unless otherwise stated). rMpAOX elution was carried out with buffer A added of 2.5 mM desthiobiontin. Protein purification was followed by SDS-PAGE and coomassie brilliant-blue staining[61] and western blot with anti-twin-strep (IBA GmbH, 2-1509-001) and anti-AOX[62] which was kindly provided by Prof. T. Elthon, Nebraska, USA. Protein content was estimated through the Bradford method[63] and bovine serum albumin as standard.

**NDH-2 expression and purification**. Expression and purification of a bacterial type II NADH dehydrogenase (NDH-2) from *Caldalkalibacillus thermarum* was performed as described elsewhere[64]. The plasmid pTRCndh2 was kindly donated by Prof. Greg Cook (Otago University, New Zealand). In summary, overexpression was carried out in *E. coli* C41 for 4 h with 1 mM IPTG. Cells were harvested and lysed, the membrane fraction was collected and NDH-2 was solubilised with 2% *n*-octyl-β-D-glucopyranoside (OG). NDH-2 was purified through metal-affinity chromatography[64].

**Self-assembled proteoliposome preparation**. Proteoliposomes (PL) were prepared as previously described[44,45], except that NDH-2 was used instead of complex I and 0.5% DDM was used instead of OG. For IC$_{50}$ measurements, PLs were prepared with 5 μg of rMpAOX and 500 μg of NDH-2. For kinetic studies, PLs were prepared with 50 μg rMpAOX and no NDH-2—in this case, pure NDH-2 was added directly into the enzymatic reaction mixture at a saturating concentration. Briefly, lipids and Q$_{10}$ stocks in chloroform were mixed in a glass vial and the chloroform dried under a nitrogen stream. The lipids were hydrated and extruded through a 100 nm NanoSizer MINI Liposome Extruder (T&T Scientific) followed by the addition of protein. After treatment with the Bio-Beads SM-2 Resin (Bio-Rad), proteoliposomes were harvested by ultracentrifugation and resuspended in 150 μL of buffer. Enzymatic assays were performed on the same day.

**Enzymatic activity measurements**. Oxygen uptake from *E. coli* membrane-bound rMpAOX was measured in an Oxygraph-2k (Oroboros Inc.) high-resolution respirometer in 65 mM MOPS pH 7.5 with 1.25 mM NADH and 0.1 mg mL$^{-1}$ potassium cyanide. Respiration rates were recorded before and after the addition of putative AOX activators, such as 5′-ADP, 5′-AMP, 5′-GDP, 5′-GMP, glyoxylate, oxaloacetate, pyruvate or succinate (1 mM each unless stated otherwise). NADH oxidation measurements of membrane-bound rMpAOX were performed spectrophotometrically ($\varepsilon_{340} = 6.22$ mM$^{-1}$ cm$^{-1}$) with 0.1 mg mL$^{-1}$ KCN, 1 mM GMP and 300 μM NADH in 200 μL reactions. Data were acquired with the Multiskan FC Microplate Photometer (ThermoFisher). For the pure rMpAOX, Q$_1$H$_2$ oxidation was measured spectrophotometrically ($\varepsilon_{284} = 14.1$ mM$^{-1}$ cm$^{-1}$) with 100 ng enzyme in 50 μL of 20 mM tris-HCl pH 7.5 and 0.0125% octaethylene glycol monododecyl ether (C12E8) in 384-well microplates (CLARIOstar Plus Microplate Reader, BMG LABTECH). Absorbance readings were taken periodically and a blank reactions (substrate without enzyme) were included in every assay and were used to determine and subtract the autoxidation rate (background). Proteoliposomes were assayed as described for the *E. coli* membrane fraction except that no KCN was included and 0.5 μL of the prep (~50 ng rMpAOX) were added into the reaction. When tested, inhibitors were preincubated with the enzyme in reaction buffer for 2–5 min before starting the reaction by the addition of substrate. The path length (i.e., the height of the reaction medium inside the well) was calculated based on the reaction volume and the well cross-section area informed by the supplier. GraphPad Prism 7.0 (GraphPad Software Inc.) was used for function fitting.

**MpAOX sequence alignment and structural modelling**. The amino acid sequence of AOXs used here are from *Trypanosoma brucei* (GenBank ID BAB72245.1), *Moniliophthora perniciosa* (ABN09948.3), *Septoria tritici* (XP_003851917.1), *Neurospora crassa* (XP_962086.1), *Aspergillus niger* (BAA32033.2), *Arabidopsis thaliana* (NP_188876.1), *Oryza sativa* (XP_015635413.1) and *Zea mays* (NP_001105180.1). The sequence alignment was performed with the MUSCLE algorithm built in JalView[65] with default parameters. The MpAOX protein sequence was used as input in the SWISS-MODEL protein structure modelling server[66] and TAO crystal structures PDB ID 3vv9 and 3w54[42] were selected as templates. The quality of the output model was checked within parameters provided by SWISS-MODEL server. The MpAOX model and TAO structure 3w54 were overlaid with PyMOL molecular viewer 2.3.3 (Schrödinger, LLC) and visually inspected, in particular residues that form the active site in TAO and residues with known functions[6]. The volume of the ligand-binding pocket was calculated with the KVfinder plugin[67] with "probe in" and "probe out" radii of 1.1 and 5 Å, respectively. The secondary structure prediction and solvent accessibility prediction were performed with NetSurfP 2.0[68], Jpred 4[69], PSIPRED[70] and PredictProtein[71].

**Statistics and reproducibility**. Measurements were performed in technical triplicate and each experiment was performed three independent times (e.g. distinct protein purification batches) unless otherwise stated and similar results were obtained each time. No data points were excluded from the analyses. Data fitting and plotting were performed with GraphPad Prism 7.0 (GraphPad Software Inc.).

**Reporting summary**. Further information on research design is available in the Nature Research Reporting Summary linked to this article.

## Data availability

All data used in this study are included within the main text or provided in the supplementary files. Source data for graphs/charts in the main figures is available in Supplementary Data 1. Any additional information and biological material described in the study is available from the corresponding authors upon request.

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

## Acknowledgements

This work was supported by São Paulo Research Foundation (FAPESP) through research Grants 2015/07653-5 and 2016/10498-4, and scholarships 2014/15339-6, 2017/12852-2 and 2017/13319-6. This work was also supported by the National Council for Scientific and Technological Development (CNPq) Grants 475535/2013-8 and 424949/2018-0, and scholarship 142358/2014-2. Work in ALM's laboratory was supported by BBSRC (BB/NO10051) and the University of Sussex Strategic Development Fund.

## Author contributions

M.R.O.B. helped in experimental design, performed experiments, analysed data, made figures, and wrote and reviewed the manuscript. A.C., L.Y. helped in experimental design, performed experiments, analysed data and reviewed the manuscript. R.M.B. performed experiments and reviewed the manuscript. A.T.C. helped in experimental design, analysed data and provided supervision. G.A.G.P. conceived the project, provided funding and supervision, administered the project and reviewed the manuscript. A.L.M. conceived the project, helped in experimental design, provided funding and supervision, administered the project and reviewed the manuscript.

## Competing interests

The authors declare no competing interests.
