## [Peer Review file · Communications Biology]

Reviewers' comments:

Reviewer #1 (Remarks to the Author):

This is an interesting study on the structure function relationships of an AOX protein from a fungus that is a danger to Cocoa crops. The experiments have been very well conducted and the results are presented well. I have no issue with the experiments or the results per se. Overall this is a solid piece of research building on previous studies of the AOX enzyme from a number of species.

Will the results lead to the development of a specific fungicide for this pest? Possibly, but I am not optimistic about that. A key issue with attempts to use mitochondrial electron transport components as targets for pests is that they are key enzymes for the plant as well and, in the case of AOX, are important for the plant's response to environmental stress. The authors mention very briefly this problem but gloss over it. At the very least, I think they need to show the relative effects if the inhibitors tested on plant AOX enzymes also, preferably AOX from the cocoa plant. If there are clear differences then I think they are onto something important.

The manuscript needs to be carefully edited for correct English expression.

Reviewer #2 (Remarks to the Author):

This work shows a biochemical characterization of recombinant *M. perniciosus* AOX. This fungus is the causal agent of witches' broom disease of cocoa, a very important product from an economical point of view. ADP, AMP and GMP activated the membrane-associated rMrAOX. Dose-response curves for six inhibitors were obtained using the membrane-bound rMrAOX. AF and CB were the most potent inhibitors and SHAM did not affect rMrAOX at the concentrations tested. To explain the differences in inhibitor potency between rMrAOX and TAO a structural model of MpAOX was created based on the crystal structure of TAO (PDB ID 3w54) and some key residues explain the change in the affinity of AC. It was found Michaelis-Menten kinetics for purified (Q1H2) and reconstituted in liposomes (Q10H2) rMrAOX, but with large differences in the V_{max} .

The manuscript can be published after major corrections. Major and minor questions are indicated below.

1. The work showed that ADP, AMP and GMP are activators of the membrane-bound rMrAOX. Next, a dose-response curve for GMP was obtained, but not with ADP and AMP, even though a) the concentrations of ADP and AMP are frequently larger than the concentration of GMP (Bennett, et. al., Absolute metabolite concentrations and implied enzyme active site occupancy in *Escherichia coli*. Nat Chem Biol. 2009 5(8):593-9. doi: 10.1038/nchembio.186.), and b) changes in the cytosolic and mitochondrial concentrations ADP and AMP might be quite large in the presence of cyanide, a condition that activates fungal AOXs. So, it is likely that regulation of MrAOX –and probably most of the fungal AOXs- relies more on AMP than GMP. From a physiological point of view, it would be important to get the kinetics of rMrAOX activation by ADP and AMP. Are the V_{max} and K_a the same for the three nucleotides?

2. It is stated that in figure legend of figure 1 "the continuous line is the best fit for a 4-parameter sigmoid function". Does this mean that activation by GMP did not follow Michaelis-Menten (MM) kinetics? Sigmoid curves for MM kinetics are obtained when data is displayed in a semi-log plot.

3. To explain the difference in V_{max} between the purified rMrAOX and the rMrAOX incorporated in liposomes it was assumed that the results obtained with Q1H2 are inaccurate because of the low CMC of Q1H2 (150 μ M), resulting in limited availability of free Q1H2 as the total concentration of Q1H2 increases. However, fig3c shows a good fit of the MM equation to the data up to 120 μ M, or even 75 μ M, both concentrations below Q1H2 CMC. Furthermore, if it is assumed that the real V_{max} is close to 100 μ mol/min/mg, then most of the point in fig 3c should fall on a straight line because the concentrations of Q1H2 used are far below the K_m .

4. The activity of membrane-bound rMrAOX is 1,338 nmol Q min⁻¹ mg total protein⁻¹. Since the anti-AOX antibody and the purified rMrAOX are available, it is possible

to prepare a standard curve using the purified rMrAOX (Western blot), and determine the contribution of the AOX to the total protein and calculate the specific activity of rMrAOX per mg AOX in both *E. coli* membranes and *M. perniciosus* mitochondria. According to the gel, it can be guessed that AOX makes a third of the total membrane protein, and thus the activity should increase to 4 micromol Q min⁻¹ mg AOX⁻¹.

5. Is it possible that the high V_{max} (close to 100 micromol Q min⁻¹ mg AOX⁻¹) might be due to a miscalculation? Obtaining the highest slope from supplementary figure S3b (-0.0313 min⁻¹), and assuming a 0.5 cm for the light path length (absorbance of 300 μM NADH solution should be 1.86 in 1 cm light path length cuvette), and 200 μL of reaction mix, several specific activities can be calculated depending on the amount of protein in those 200 μL. Since this value is not established, one can start with 50 μg/200 μL (lines 377-379):

$(0.0313/\text{min}) \cdot (1\text{mM} \cdot \text{cm} / 6.22) \cdot (1000 \text{ nmol} / (\text{mL} \cdot 1\text{mM})) \cdot (0.2 \text{ mL} / 0.5 \text{ cm}) \cdot (1 / 0.05\text{mg}) = 0.04 \text{ } \mu\text{mol} / (\text{min} \cdot \text{mg})$

With 5 μg/200 μL, the result is 0.4 μmol/(min·mg) which is close to the V_{max} of purified rMrAOX. With 0.02 μg (20 ng), the result is 100 μmol/(min·mg). Is this the amount of protein used per assay?

The final volume and the amount of protein used in each assay should be written in the figure legends.

6. Lines 130-131: is not clear the end of the sentence "This indicates that AC is a much weaker ligand for MpAOX – m a valuable piece of information for the design of compounds targeted at *M. perniciosus*"

24th March 2020

To the reviewers

Dear Sir/Madam,

We are pleased to submit a revised and improved version of the manuscript entitled **Biochemical characterization and inhibition of the alternative oxidase enzyme from the fungal phytopathogen *Moniliophthora perniciosa***. We appreciate the efforts and insightful comments from the editor and the reviewers, and we have strived to answer all concerns raised. We have included new experimental data and have amended the text to reflect those and other changes. Points in the original manuscript that were not clear or lacking information were amended. The point-by-point answers with references to the revised manuscript are found below.

Reviewer #1:

1.1. *Will the results lead to the development of a specific fungicide for this pest? Possibly, but I am not optimistic about that. A key issue with attempts to use mitochondrial electron transport components as targets for pests is that they are key enzymes for the plant as well and, in the case of AOX, are important for the plant's response to environmental stress. The authors mention very briefly this problem but gloss over it. At the very least, I think they need to show the relative effects if the inhibitors tested on plant AOX enzymes also, preferably AOX from the cocoa plant. If there are clear differences then I think they are onto something important.*

R: The authors are aware of possible limitations of using electron transport chain inhibitors and the possible inhibition of the plant AOX as a side-effect of a fungicide treatment. Indeed, our group has begun investigating their effect on plant AOXs, such as from *A. thaliana* and *Sauromatum guttatum* which will be presented in future publications. However, we believe that answering those questions is outside of the present work's scope and we have made the conscious choice of only mentioning this aspect briefly in the Discussion section. We wish not to digress from this manuscript's contributions which in summary are 1) the purification and biochemical characterisation of a fungal AOX, 2) the analysis of structure-activity relationships of fungal AOX inhibitors, and 3) the importance of the phospholipid membrane for the determination of true AOX kinetic parameters. Nonetheless, we take this opportunity to present some arguments in favour of using AOX inhibitors as antifungal agents which can be found below.

The effect of a small molecule in an organism is not dictated solely by the affinity towards the molecular target (potency), but also by a myriad of properties and interactions within the biological system which are broadly studied in the fields of Pharmacokinetics and Pharmacodynamics. In other words, it is possible that with the correct dosage and timing of application the side-effects of AOX inhibitors in plants will not be as severe when compared to their lethality against fungal cells. This

could happen due to differences in the metabolism and physiology between both organisms, the occurrence of physical barriers in plants such as the bark or the cuticle, or that fungi might be under greater physiological stress than plants during the beginning of the infection. In fact, Barsottini et al. (2018) have demonstrated that the chemical inhibition of *M. perniciosa* AOX (without the combination of a main respiration inhibitor) prevents spore germination and disease development in cocoa plantlets [Pest Manag Sci. 2019, 75(5):1295-1303].

Furthermore, there are already plenty of respiration inhibitors successfully used as fungicides in the market. In 2017, 236 active ingredients were commercialised from which 54 (23%) are respiration inhibitors. Forty-four are complex II or III inhibitors which occur in virtually every eukaryotic cell and yet this did not prohibit the clinical or agricultural use of those products [Fungicide Resistance Action Committee, 2020]. Moreover, strobilurins – the most successful class of complex III inhibitors – have been safely used as crop protection agents since the 1990's. Strobilurins alone comprised 20% of global fungicide sales in the following decade and azoxystrobin was the best-selling fungicide only 4 years after being launched [Pest Manag Sci. 2003, 59(5):499-511; Fungicides 2010, Odile Carisse, IntechOpen, Ch 10:203-220].

1.2. *The manuscript needs to be carefully edited for correct English expression.*

R: The manuscript has been thoroughly revised by our three native English authors.

Reviewer #2:

2.1a. *The work showed that ADP, AMP and GMP are activators of the membrane-bound rMrAOX. Next, a dose-response curve for GMP was obtained, but not with ADP and AMP, even though a) the concentrations of ADP and AMP are frequently larger than the concentration of GMP (Bennett, et. al., Absolute metabolite concentrations and implied enzyme active site occupancy in Escherichia coli. Nat Chem Biol. 2009 5(8):593-9. doi: 10.1038/nchembio.186.), and b) changes in the cytosolic and mitochondrial concentrations ADP and AMP might be quite large in the presence of cyanide, a condition that activates fungal AOXs. So, it is likely that regulation of MrAOX –and probably most of the fungal AOXs- relies more on AMP than GMP. From a physiological point of view, it would be important to get the kinetics of rMrAOX activation by ADP and AMP.*

R: We now present dose-response curves for ADP and AMP, and we have included GDP in the panel of tested activators which had no effect on rMpAOX at 1 mM. Results, Discussion and Methods sections have been amended accordingly (**lines 105-110, lines 267-276, line 352 and Fig. 1**).

2.1b. *Are the Vmax and Ka the same for the three nucleotides?*

R: We have now determined that the activation level of AMP is similar to that of GMP at the highest concentrations tested. ADP did not reach a plateau - therefore V_{Max} could not be determined but it is possibly as high as AMP and GMP at high enough concentrations. The EC_{50} values for GMP, AMP and ADP are 0.18 mM, 0.91 mM and ca. 6.5 mM and the Results and Discussion sections have been updated accordingly (**lines 105-110, lines 267-276 and Fig. 1B**). Because those activators have no effect on the purified rMpAOX it was not possible to determine the dissociation constant, K_a . I.e., the purified rMpAOX V_{Max} and K_M remain unchanged in the presence of those nucleotides.

2.2. It is stated that in figure legend of figure 1 “the continuous line is the best fit for a 4-parameter sigmoid function”. Does this mean that activation by GMP did not follow Michaelis-Menten (MM) kinetics? Sigmoid curves for MM kinetics are obtained when data is displayed in a semi-log plot.

R: The data points in Fig. 1B create hyperbolae in a linear plot and a sigmoid curves in a semi-log plot. We have chosen the well-established approach of using the semi-log plot and a sigmoid function to determine the activator concentration necessary for half-maximal activation (EC_{50}) which has the additional advantage of making it easier to visually gauge the EC_{50} of different activators on the same graph. Due to limitations imposed by the *E. coli* membrane system – such as undefined substrate concentration and the fact that rMpAOX cannot reach maximal rate – it is not possible to properly model the enzymatic kinetic (the reader is referred to the comment 2.4 for more details). The purified rMpAOX displayed MM behaviour in the presence of the activators, albeit without any change in K_M and V_{Max} .

2.3. To explain the difference in V_{max} between the purified rMrAOX and the rMrAOX incorporated in liposomes it was assumed that the results obtained with Q1H2 are inaccurate because of the low CMC of Q1H2 (150 μM), resulting in limited availability of free Q1H2 as the total concentration of Q1H2 increases. However, fig3c shows a good fit of the MM equation to the data up to 120 μM , or even 75 μM , both concentrations below Q1H2 CMC. Furthermore, if it is assumed that the real V_{max} is close to 100 $\mu\text{mol}/\text{min}/\text{mg}$, then most of the point in fig 3c should fall on a straight line because the concentrations of Q1H2 used are far below the K_m .

R: The reviewer’s argument is correct. Indeed we have initially hypothesized that the limited Q_1H_2 solubility in aqueous medium might have explained the low catalytic rates obtained with this substrate. This matter has been explored before by Hoefnagel et al. (1997) and Kido et al. (2010) who have demonstrated that AOXs from *Arum italicum* and *Trypanosoma brucei* show linear dependencies with respect to Q_1H_2 concentration in the absence of solubilizing agents (e.g., detergents) [Plant Physiol. 1997, 11(5):1145-1153; Biochim. Biophys. Acta 2010, 1797:443-450]. Since we have added C12E8 in the enzymatic reaction medium it is expected that Q_1H_2 will be fully soluble up to 600 μM . However, Hoefnagel et al. also report that Q_1H_2 micelle formation is time-dependent. Since we are using a different experimental method (microtiter plates vs. quartz cuvettes) which undoubtedly takes longer to setup, we could not confidently rule out the occurrence

of micelles. In other words, the apparent good fit for the MM model could have been spurious and in theory explained by intermediate states between fully solubilized Q_1H_2 (at low concentrations) and at least partially aggregated Q_1H_2 (at higher concentrations). In order to investigate this hypothesis, additional experiments were carried out and our conclusion was that, in fact, it is unlikely that low substrate solubility explains the low rMpAOX catalytic rates. Instead, the short isoprenoid tail of Q_1H_2 a more plausible reason. For instance, there was no difference in Q_1H_2 oxidation in the presence of phospholipids which could facilitate the diffusion of the substrate into the rMpAOX catalytic site (**Sup. Fig S6**) and the oxidation rate of $180 \mu M Q_{10}H_2$ is approximately 26 times faster than that of $180 \mu M Q_1H_2$ ($9.3 \mu mol \text{ min}^{-1} \text{ mg}^{-1}$ and $0.36 \mu mol \text{ min}^{-1} \text{ mg}^{-1}$, respectively) – which is in line with the reviewer's argument. We have modified the Discussion section in order to improve the clarity on this subject (**lines 238-249 and supplementary Fig. S1**).

2.4. The activity of membrane-bound rMrAOX is $1,338 \text{ nmol Q min}^{-1} \text{ mg total protein}^{-1}$. Since the anti-AOX antibody and the purified rMrAOX are available, it is possible to prepare a standard curve using the purified rMrAOX (Western blot), and determine the contribution of the AOX to the total protein and calculate the specific activity of rMrAOX per mg AOX in both *E. coli* membranes and *M. perniciosus* mitochondria. According to the gel, it can be guessed that AOX makes a third of the total membrane protein, and thus the activity should increase to $4 \text{ micromol Q min}^{-1} \text{ mg AOX}^{-1}$.

R: Yes, we had indeed quantified the amount of rMpAOX bound to the *E. coli* membrane through western blot using known amounts of purified protein as a standard which is now being presented as suggested by the reviewer. We had not originally shown that data because, as later observed, rMpAOX is not rate-limiting for the electron flux in the *E. coli* membrane. As seen in the new **Supplementary Fig. S1** the mean rMpAOX specific activity is $5 \mu mol O_2 \text{ min}^{-1} \text{ mg AOX}^{-1}$ which is equivalent to $10 \mu mol Q \text{ min}^{-1} \text{ mg AOX}^{-1}$. This corresponds to only 16% of the V_{Max} seen in the proteoliposome system and indicates that the overexpressed rMpAOX activity is greatly limited by the capacity of the bacterial NADH dehydrogenases to reduce the ubiquinone pool and provide rMpAOX with sufficient amounts of substrate. We have also observed a large variation in rMpAOX content between different *E. coli* membrane preparations (between 6% and 31% of the total protein content) which leads to an equally variable calculated specific activity of rMpAOX. Overall those facts limit any conclusion that one might make about rMpAOX kinetics in the *E. coli* membrane. This was in fact one of the reasons which made us push forward with the development of the proteoliposome system. The manuscript now includes this information (**line 105, lines 260-264 and supplementary Fig. S1**). Unfortunately, we no longer have access to *M. perniciosus* to carry out the experiments necessary for determining the MpAOX specific activity in isolated mitochondria.

2.5. Is it possible that the high V_{max} (close to $100 \text{ micromol Q min}^{-1} \text{ mg AOX}^{-1}$) might be due to a miscalculation? Obtaining the highest slope from supplementary figure S3b (-0.0313 min^{-1}), and assuming a 0.5 cm for the light path length (absorbance of 300 .iM NADH solution should be 1.86 in

1 cm light path length cuvette), and 200 .iL of reaction mix, several specific activities can be calculated depending on the amount of protein in those 200 uL. Since this value is not established, one can start with 50 .ig/200 .iL (lines 377-379): $(0.0313/\text{min}) \cdot (1\text{mM} \cdot \text{cm}/6.22) \cdot (1000 \text{ nmol}/(\text{mL} \cdot 1\text{mM})) \cdot (0.2 \text{ mL}/0.5 \text{ cm}) \cdot (1/0.05\text{mg}) = 0.04 \text{ .imol}/(\text{min} \cdot \text{mg})$. With 5 .ig/200 uL, the result is 0.4 .imol/(min*mg) which is close to the Vmax of purified rMrAOX. With 0.02 .ig (20 ng), the result is 100 .imol/(min*mg). Is this the amount of protein used per assay? The final volume and the amount of protein used in each assay should be written in the figure legends.

R: The representative curves originally shown in figure S3b for the PL system were taken from the inhibitor dose-response assay and not the substrate titration assay and we apologize for the lack of clarity. For the dose-response assay NDH-2 is added in excess alongside rMpAOX during sample preparation and the exact concentration of rMpAOX cannot be easily determined, therefore the specific activity was not calculated. In order to comply with the reviewer's suggestion we are now presenting an updated **Supplementary Fig. S5** with rates corresponding to the Q₁₀ titration assay and known amounts of rMpAOX. However, during preparation of that figure we noticed an error in our original calculations which led to incorrect rMpAOX K_M and V_{Max} values. The correct values are $0.96 \pm 0.76 \text{ mM}$ and $59.4 \pm 6.6 \mu\text{mol Q}_{10} \text{ min}^{-1} \text{ mg}^{-1}$ and the main text and associated figure have been corrected (**lines 193-194, lines 235-237 and Fig 4a**). The correct calculation with one data point is shown below so the reviewer may check it. We note that this error does not affect our initial conclusions with respect to the importance of providing a physiologically-relevant environment to measure true kinetic parameters of AOX, such as a the phospholipid membrane and the natural substrate Q₁₀, since there is still a dramatic increase in enzymatic activity after incorporation of rMpAOX in proteoliposomes with Q₁₀. We have re-checked the other results originally presented and they are correct.

The absorbance change rate obtained with the highest concentration of Q₁₀ was $0.066 \text{ AU min}^{-1}$. With a path length of 0.63 cm, a reaction volume of 0.2 mL, an extinction coefficient of $6.22 \text{ mM NADH}^{-1} \text{ cm}^{-1}$ and 63.4 ng rMpAOX in the mixture – one can calculate that the resulting NADH oxidation rate is $52.3 \mu\text{mol min}^{-1} \text{ mg}^{-1}$ which is equivalent to the same rate of Q₁₀ oxidation.

2.6. Lines 130-131: is not clear the end of the sentence “This indicates that AC is a much weaker ligand for MpAOX – m a valuable piece of information for the design of compounds targeted at *M. perniciosa*”.

R: The typographic error in that sentence has been amended which now reads “This indicates that AC is a much weaker ligand for MpAOX – a valuable piece of information for the design of compounds targeted at *M. perniciosa*” (**lines 127-128**).

Sincerely,

Professor Anthony Moore.

Reviewer #1 (Remarks to the Author):

Notwithstanding the authors' response, I still think they need to show the effect of inhibitors on plant AOX for this study to be published. They mention that they have done these studies and I think they need to show the results.

Reviewer #2 (Remarks to the Author):

I agree with the arguments and the changes the authors made to the manuscript. I recommend the publication of this work. In the new version I found a typo error in line 67: "as seen in *Ustilago maydis*, 14, *Sclerotinia sclerotiorum*", there is a misplaced comma.